# Association of SNPs in the PAI1 Gene with Disease Recurrence and Clinical Outcome in Bladder Cancer

**DOI:** 10.3390/ijms24054943

**Published:** 2023-03-03

**Authors:** Kaoru Murakami, Hideki Furuya, Kanani Hokutan, Steve Goodison, Ian Pagano, Runpu Chen, Cheng-Huang Shen, Michael W. Y. Chan, Chi Fai Ng, Takashi Kobayashi, Osamu Ogawa, Makito Miyake, Mark Thornquist, Yoshiko Shimizu, Kazukuni Hayashi, Zhangwei Wang, Herbert Yu, Charles J. Rosser

**Affiliations:** 1Samuel Oschin Comprehensive Cancer Institute, Cedars-Sinai Medical Center, Los Angeles, CA 90048, USA; 2Clinical and Translational Research Program, University of Hawaii Cancer Center, Honolulu, HI 96813, USA; 3Department of Molecular Biosciences and Bioengineering, University of Hawaii at Manoa, Honolulu, HI 96822, USA; 4Department of Quantitative Health Sciences, Mayo Clinic, Jacksonville, FL 32224, USA; 5Population Sciences in the Pacific Program, University of Hawaii Cancer Center, Honolulu, HI 96813, USA; 6Department of Microbiology and Immunology, The State University of New York at Buffalo, Buffalo, NY 14260, USA; 7Department of Urology, Ditmanson Medical Foundation Chia-Yi Christian Hospital, Chiayi 600, Taiwan; 8Department of Biomedical Sciences, National Chung Cheng University, Chia-Yi 621, Taiwan; 9SH Ho Urology Centre, Department of Surgery, The Chinese University of Hong Kong, Hong Kong; 10Department of Urology, Kyoto University Graduate School of Medicine, Kyoto 606-8507, Japan; 11Department of Urology, Nara Medical University, Nara 6348522, Japan; 12Fred Hutchinson Cancer Research Center, Seattle, WA 98109, USA

**Keywords:** plasminogen activator inhibitor-1, bladder cancer, 3′ UTR SNP, recurrence, survival, apoptosis, Claspin

## Abstract

**Simple Summary:**

Bladder cancer (BCa) is one of the most common cancer types worldwide and is characterized by a high rate of recurrence. Previous studies have demonstrated that plasminogen activator inhibitor-1 (PAI1) plays an important role in bladder cancer development. The aim of our retrospective study was to assess the effects of *PAI1* mutational status in bladder tumors. In this study, we evaluated the mutational status of PAI1 in a series of independent cohorts, comprised of a total of 660 subjects. We identified two clinically relevant 3′ untranslated region (UTR) single nucleotide polymorphisms (SNPs) in *PAI1*. Caucasian patients with at least one of the SNPs had a high risk of recurrence and shorter survival. Additional studies using cell lines demonstrated that one SNP increased the anti-apoptotic effect of PAI1, and another lost the control of cancer cellular growth. This study underscores the relevance and influence of these SNPs in bladder cancer.

**Abstract:**

Purpose: Bladder cancer (BCa) is one of the most common cancer types worldwide and is characterized by a high rate of recurrence. In previous studies, we and others have described the functional influence of plasminogen activator inhibitor-1 (PAI1) in bladder cancer development. While polymorphisms in *PAI1* have been associated with increased risk and worsened prognosis in some cancers, the mutational status of *PAI1* in human bladder tumors has not been well defined. Methods: In this study, we evaluated the mutational status of PAI1 in a series of independent cohorts, comprised of a total of 660 subjects. Results: Sequencing analyses identified two clinically relevant 3′ untranslated region (UTR) single nucleotide polymorphisms (SNPs) in *PAI1* (rs7242; rs1050813). Somatic SNP rs7242 was present in human BCa cohorts (overall incidence of 72%; 62% in Caucasians and 72% in Asians). In contrast, the overall incidence of germline SNP rs1050813 was 18% (39% in Caucasians and 6% in Asians). Furthermore, Caucasian patients with at least one of the described SNPs had worse recurrence-free survival and overall survival (*p* = 0.03 and *p* = 0.03, respectively). In vitro functional studies demonstrated that SNP rs7242 increased the anti-apoptotic effect of PAI1, and SNP rs1050813 was related to a loss of contact inhibition associated with cellular proliferation when compared to wild type. Conclusion: Further investigation of the prevalence and potential downstream influence of these SNPs in bladder cancer is warranted.

## 1. Introduction

Bladder cancer (BCa) is the sixth most common and the seventh most lethal cancer in the US, with an estimated 81,000 new cases and 17,000 deaths in 2022 [1,2]. Although BCa incidence continues to decline annually, the death rate remains stable. BCa is categorized into two subclassifications: non-muscle-invasive (NMIBC) and muscle-invasive (MIBC). Approximately 70–80% of bladder cancers are diagnosed as NMIBC, with the remaining 20% of patients presented with MIBC or advanced disease [3]. Although NMIBC can be treated surgically via transurethral resection (TUR) and has an excellent 5-year survival rate, 50–70% of NMIBC patients will have a recurrence within 5 years after TUR [4]. MIBC has a poor survival rate (50% at 5 years), and those with metastatic disease have a 5-year survival rate of 10–33% [3].

In a series of previous studies [5,6,7,8], we have identified multiplex panels of biomarkers that are being developed into clinical tests for the non-invasive diagnosis and prognosis of BCa [9,10]. Included in these biomarker panels is plasminogen activator inhibitor-1 (PAI1, also known as SERPINE1). The PAI1 gene is located on chromosome 7q22.1 and contains nine exons (CDS- exon 2–9). Translated PAI1 is a globular protein composed of three beta-sheets (A, B, C), nine alpha-helical domains (A-I) and a reactive center loop (RCL) [11]. The canonical role of PAI1 is as the principal inhibitor of urokinase plasminogen activator (uPA) and tissue plasminogen activator (tPA), which regulate plasminogen cleavage [12] and the maintenance of blood clots. In the oncology literature, several studies have reported that elevated levels of tumoral PAI1 are associated with poor clinical outcomes [13,14,15,16]. Molecular studies by us and others show that besides its role in coagulation, PAI1 activity can impact multiple pathways involved in cancer development, including cellular proliferation, adhesion, migration, and invasion, and angiogenesis [17,18].

Cancer is a genetic disease driven by heritable or somatic mutations, and a number of gene alterations have been identified in BCa. For example, *FGFR3* mutations are often present in NMIBC and inactivating *TP53* mutations are prevalent in MIBC [19,20]. Polymorphisms in *PAI1* have been identified in a number of cancer types, including breast, testicular, and gastric cancers [21,22,23], but their prevalence and potential mechanism of action in BCa has not been investigated. In this study, we screened a series of BCa sample sets for *PAI1* genetic alterations by PCR and bidirectional sequencing and evaluated the potential correlation with clinicopathological characteristics of patients with BCa. The presence of two 3′ untranslated region (UTR) single nucleotide polymorphisms (SNPs; rs7242 and rs1050813) were found to be correlated with a worse clinical outcome in Caucasian patients. In a series of in vitro cell-based studies, the impact of these *PAI1* SNPs was investigated and potential mechanisms of action were identified.

## 2. Results

Genetic alterations of the PAI1 gene were present in exons 2 and 9 in bladder cancer cell lines.

We profiled 12 human urothelial cell lines to evaluate the prevalence of *PAI1* genetic alterations. The cell lines consisted of benign, transitional cell papilloma, and urothelial carcinoma (UC) grades I–IV. The gene promoter and exon regions of *PAI1* were analyzed using amplification by polymerase chain reaction (PCR) and direct sequencing. Point mutations and insertions were found in nine cell lines (9 of 12; 75%). No alterations were detected in the cell lines UROtsa (benign), RT112 (UC, grade II), or T24 (UC, grade III). Genetic alterations in *PAI1* are summarized in Appendix A. In bladder cell lines, genetic alternations are found only in exons 2 and 9. Therefore, we decided to focus only on exons 2 and 9 in the subsequent human BCa cohorts 1–3.

### 2.1. Genetic Alterations of the PAI1 Gene in Human BCa Specimens (Cohort 1)

The next cohort evaluated comprised 92 DNA samples extracted from urine cell pellets (case = 69; control = 23) obtained from the NCI’s Early Detection Research Network (EDRN). Notably, the two alterations most reported to be present within *PAI1* (A15T, rs6092; V17I, rs6090) were not detected in the human clinical sample set. However, the most common *PAI1* genetic alterations observed in the cell lines (rs11178; rs7242; rs41423845) were also detected in the human samples, albeit at lower frequencies. Furthermore, we detected three additional SNPs located within *PAI1* exon 9 in three (rs1050766; 3.3%), nine (rs1050813; 9.8%), and six (rs1050955; 6.5%) patient cell pellet samples which were not identified in the cell lines. All detected alterations are listed in Appendix A.

To determine the prevalence of *PAI1* alterations in BCa, we compared the presence of mutations in BCa cases versus controls. Mutations in *PAI1* exon 9 showed significantly higher frequency in BCa than controls (64% vs. 32%, *p* = 0.028, Fischer’s exact test). When comparing the number of polymorphisms per sample in this exon (0–6 polymorphisms), the results suggest that a polymorphism in *PAI1* exon 9 is highly associated with BCa (*p* = 0.015, Wilcoxon rank sum test). Based on these results, we focused our subsequent studies on the analysis of *PAI1* exon 9.

### 2.2. PAI1 Mutations Correlate with Clinicopathological Characteristics of Patients with BCa (Cohort 2—Discovery Cohort)

To examine the clinical impact of genetic variations on clinicopathological outcomes, we analyzed 73 annotated fresh frozen BCa tumor tissues (Table 1). Similar to Cohort 1, we found several major genetic alterations in *PAI1* exon 9 (Appendix A). Twelve point mutations within *PAI1* exon 9 were identified, of which, six were identified in Cohort 1: rs11178 (14.1% in Cohort 1, 19.2% in Cohort 2), rs1050766 (3.3% in Cohort 1, 2.6% in Cohort 2), rs7242 (10.9% in Cohort 1, 48.7% in Cohort 2), rs1050813 (9.8% in Cohort 1, 6.4% in Cohort 2), rs2227716 (2.2% in Cohort 1, 5.1% in Cohort 2), and rs1050955 (6.5% in Cohort 1, 3.8% in Cohort 2).

Associations of *PAI1* alterations with clinicopathological characteristics were analyzed with SAS 9.4 PHREG and LIFEREG time-to-event procedures (SAS Institute Inc., Cary, NC, USA). Information on age, gender, race, tumor grade, tumor stage, smoking history, disease recurrence, disease progression, and death were available for analysis (Table 2). Significant associations between age and genetic alterations with BCa recurrence were noted (Table 2). Patients >65 years were associated with a shorter median time to recurrence as estimated from a parametric model (*p* = 0.011). Furthermore, patients harboring SNPs in *PAI1* exon 9 (rs7242, *p* = 0.009; rs1050813, *p* = 0.009) had a shorter median time to recurrence (Table 2).

In order to determine whether these tumor recurrence-associated genetic alterations were somatic or germ line, we screened DNA extracted from patients’ matched buffy coat samples. Sequence analysis determined that SNP rs7242 was present in bladder tumor tissue, but not in matched buffy coat sample DNA, indicating a somatic mutation. Conversely, SNP rs1050813 was present in both tumor and buffy coat samples, indicating an inherited germ line mutation. The haplotype for rs7242 was TT (WT—21/62, 34%), TG (Heterozygote—27/62, 44%), and GG (Homozygote—14/61 23%), while the haplotype for rs1050813 was GG (WT—52/73, 72%), GA (Heterozygote 20/73, 27%), and AA (Homozygote—1/72, 1%) (Table 2), suggesting that these two loci are in a haplotype block with a high level of linkage disequilibrium (Appendix A). These two SNPs (rs7242 and rs1050813) were chosen to tag the linkage disequilibrium. The call rate of each tSNP was >90%, and the Hardy–Weinberg equilibrium test for each tSNP passed at the *p* = 0.05 level. Taken together, we focused on *PAI1* SNPs rs7242 (somatic) and rs1050813 (germ line) in subsequent studies.

### 2.3. PAI1 Mutations Correlate with Clinicopathological Characteristics of Patients with BCa in an Independent Cohort (Cohort 4—Replication Cohort)

To further evaluate the association of rs7242 and rs1050813 with clinical characteristics and outcome, and to investigate SNP association with outcomes and race, we included analysis of an independent cohort comprised of 495 patient samples; 263 from Asia, 79 from Hawaii, and 153 from mainland US. The median follow-up period for this cohort was 59.1 months. Genetic alterations of *PAI1* (rs7242 and rs1050813) were examined as described above. The allele frequency of SNP rs7242 in the replication cohort was TT—WT 27%, TG—Heterozygote 52%, and GG—homozygote 21%, and the frequency of SNP rs1050813 was GG—WT 84%, GA– Heterozygote 15%, and AA—Homozygote 1% (Table 2).

To increase analytical power, the Discovery and Replication cohorts were combined into one dataset (Table 1 and Table 2). In the pooled analysis, Caucasians, older patients (>75 years), and higher stage disease (>T1) were significantly associated with poor clinical outcome (HR, 1.881; *p* = 0.011, HR, 1.92; *p* = 0.0002 and HR, 5.23; *p* < 0.0001, respectively). We selected race and age covariates to be adjusted for in multivariate Cox proportional hazards regression analysis. We then evaluated clinical outcome data for the various genotypes of each SNP.

The overall allele frequency of SNP rs7242 for the pooled cohort was TT—WT 28%, TG—Heterozygote 51%, and GG—homozygote 21%. The frequency in the Caucasian subjects was TT WT 38%, TG—Heterozygote 43%, and GG—homozygote 19%, whereas for Asians the frequencies were TT—WT 23%, TG—Heterozygote 54%, and GG—homozygote 23% (*p* = 0.0031). In Cox proportional hazards analysis, after adjusting for age, race, and stage, for the entire cohort the TG/GG genotype was an independent indicator of recurrence-free survival, progression-free survival, and overall survival (Table 2). Compared to patients with other genotype, the adjusted hazard ratios for patients with TG/GG genotype were 1.25 for recurrence (95% CI, 0.81 to 1.93), 1.09 for progression (95% CI, 0.82 to 1.47), and 1.19 for death (95% CI, 0.87 to 1.62). Among Caucasian patients, the adjusted hazard ratios for the TG/GG genotype were 1.062 for recurrence (95% CI, 0.82 to 1.38), 1.47 for progression (95% CI, 0.90 to 2.42), and 1.35 for death (95% CI, 0.79 to 2.29). For the Asian patient subset, the adjusted hazard ratios for the TG/GG genotype were 1.06 for recurrence (95% CI, 0.76 to 1.49), 1.19 for progression (95% CI, 0.8 to 1.77), and 0.98 for death (95% CI, 0.66 to 1.47). Kaplan–Meier estimates indicated that Caucasian patients carrying the TG/GG variant allele had similar recurrence-free survival and overall survival as patients carrying the TT allele in rs7242 (*p* = 0.53 and 0.30, respectively, Figure 1). The recurrence-free survival and overall survival was not significantly different in Asians (*p* = 0.60 and 0.93, respectively, Appendix A).

The overall allelic frequency of SNP rs1050813 in the pooled cohort was GG—WT 82%, GA– Heterozygote 17%, and AA—Homozygote 1%. In Caucasians, the frequency was GG—WT 61%, GA—Heterozygote 36%, and AA—Homozygote 3%. For Asian subjects, the frequency was TT—WT 94%, TG—Heterozygote 5%, and GG—Homozygote 1% (*p* < 0.0001) (Table 1 and Table 2). Notably, males had a higher incidence of GA/AA genotype (26 vs. 11%, *p* = 0.015) compared to females. In Cox proportional hazards analysis, after adjusting for age, race, and stage, the overall GA/AA genotype was an independent indicator of recurrence-free survival, progression-free survival, and overall survival (Table 2). The adjusted hazard ratio of patients with GA/AA genotype and recurrence was 1.05 (95% CI, 0.79 to 1.39), progression was 1.05 (95% CI, 0.74 to 1.48), and death was 1.28 (95% CI, 0.89 to 1.82). In Caucasian patients, the adjusted hazard ratio of with GA/AA genotype and recurrence was 1.3 (95% CI, 0.84 to 2.02), progression was 1.16 (95% CI, 0.71 to 1.89), and death was 1.5 (95% CI, 0.91 to 2.48). In Asian patients, the adjusted hazard ratio of Asian patients with GA/AA genotype and recurrence was 1.59 (95% CI, 0.83 to 3.04), progression was 1.88 (95% CI, 0.44 to 8.01), and death was 0.88 (95% CI, 0.46 to 1.67). Kaplan–Meier estimates indicated that Caucasian patients carrying the GA/AA variant allele had shorter recurrence-free survival and overall survival than patients carrying the GG allele, but this did not reach statistical significance (*p* = 0.24 and 0.11, respectively, Figure 1). No significant differences were noted for recurrence-free survival or overall survival in Asians (*p* = 0.16 and 0.15, respectively, Appendix A).

### 2.4. Concomitant SNP rs7242 and rs1050813 Strongly Correlates with Recurrence-Free Survival and Overall Survival

The concordance of SNP rs7242 TG/GG alleles with SNP rs1050813 GA/AA alleles in the pooled cohort was 11%. The allelic frequency of SNP rs7242 TG/GG alleles with SNP rs1050813 GA/AA was greater in Caucasians than in Asians (22% vs. 3%, *p* < 0.001). However, Caucasians and Asians similarly carried at least one SNP (SNP rs7242 TG/GG or SNP rs1050813 GA/AA or both), (81% vs. 81%, *p* < 0.84). In Cox proportional hazards analysis, after adjusting for age, race, and stage, having SNP rs7242 TG/GG alleles or the SNP rs1050813 GA/AA genotype was an independent indicator of higher risks for recurrence 2.92 (95% CI, 1.4 to 6.07), progression 2.23 (95% CI, 1.22 to 4.21), and death 2.08 (95% CI, 1.08 to 4.01) in Caucasians. Kaplan–Meier estimates indicated that patients carrying the rs7242 TG/GG variant or the rs1050813 GA/AA alleles had significantly shorter recurrence-free survival and overall survival than patients carrying the rs7242 WT and rs1050813 WT alleles in Caucasians (*p* = 0.03 and 0.03, respectively, Figure 1). The recurrence-free survival and overall survival in Asians was not significantly different (*p* = 0.99 and 0.96, respectively, Appendix A).
Figure 1Kaplan–Meier survival curves for PAI1 WT vs. mutation (rs7242 alone, rs1050813 alone, or either or both rs7242 and rs1050813) in pooled cohort (Cohorts 3 and 4).
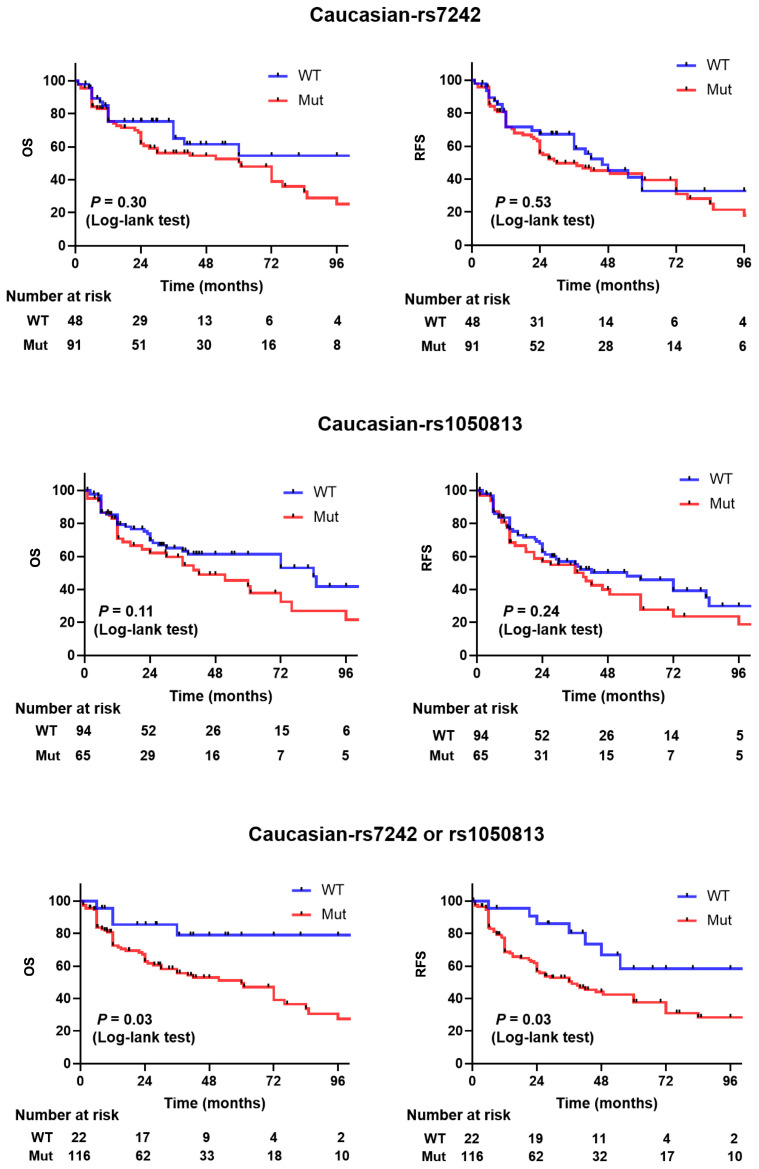


### 2.5. PAI1 mRNA Expression, RNA Structure, and miRNA Binding

We next examined whether the genetic alterations within the 3′ UTR region (including exon 9) of *PAI1* correlated with expression levels of PAI1 by quantitating mRNA expression levels in samples from Cohort 2. The presence of rs7242 or rs1050813 SNPs did not correlate with changes in mRNA expression levels (Appendix A). No evidence of changes in alternative splicing of PAI1 mRNA was observed based on transcript size.

We used RNAfold webserver to examine if the rs7242 and rs1050813 SNPs might alter the secondary structure of PAI1 mRNA. Appendix A shows mRNA structures associated with WT, rs7242 and rs1050813 transcripts. The estimated minimum free energy (MFE) value was similar between WT and rs1050813, but was lower for rs7242 than WT, suggesting a more stable structure.

Both rs7242 and rs1050813 are within the 3′ UTR of transcribed PAI1, a region that regulates mRNA stability and processing by binding protein or miRNAs. Using DIANA microT-CDS, we identified nine potential miRNAs which could bind to the 3′ UTR region of PAI1 in the presence of polymorphisms rs11178, rs2227716, rs1050955, and rs1050813. The application of STarMir verified that miR-143-3p could bind to rs1050813 in addition to PAI1 WT. The LogitProb value was lower for rs1050813 compared to WT, 0.445 vs. 0.455, respectively, suggesting a stronger interaction between miR-143-3p and PAI1 WT. However, patients harboring the variant genotype of rs1050813 displayed the same PAI1 expression levels as WT patients.

### 2.6. Differential Gene Expression Analysis

To further investigate the mechanism by which the presence of PAI1 SNPs cause reduced overall and recurrence-free survivals, we transiently expressed PAI1 WT, rs7242, and rs1050813 in UROtsa and 5637 human bladder cell lines. These cell lines were selected because they have undetectable PAI1 expression (Figure 2A), and they did not carry the SNPs (Appendix A) under study. PAI1 transfection was performed in two independent replicate experiments and expression levels were confirmed at the mRNA level (Figure 2A). Microarray profiling was used to monitor gene expression in the transfected cell series (GSE224309). Using a fold-change cutoff of 1.2, differentially expressed genes (DEGs) were identified between C125E01 (mock) and PAI1 WT, WT and rs7242, WT and rs1050813, and WT and both SNPs in UROtsa and 5637 separately. Identified DEGs were compared between UROtsa and 5637 by Venn diagram analyses (Figure 2B, Appendix A), identifying 509 common genes (C125E01 and PAI1 WT), 2350 (WT and rs7242), 5959 (WT and rs1050813), and 1341 (WT and both SNPs), respectively. The common gene lists were subjected to pathway analysis using Panther tools for each dataset (Appendix A). The identified pathways were then compared by Venn diagram analysis (Figure 2C, Appendix A), highlighting 79 common pathways, which include cell proliferation, migration, and apoptosis-related pathways, such as DNA replication (P00017), Cell cycle (P00013), Angiogenesis (P00005), Apoptosis signaling pathway (P00006), Integrin signaling pathway (P00034), p53 pathway (P00059), and Notch signaling pathway (P00045) (Pathway information is available in Appendix A.

### 2.7. Expression of PAI1 Confers Resistance to Cisplatin-Induced Apoptosis

Based on the above clinicopathological outcome and gene expression microarray results, we hypothesized PAI1 SNPs (rs7242 or rs1050813) to impact cell proliferation, migration and/or apoptosis. To address the hypothesis, we developed a set of stable transfectants (C125E01, PAI1 WT, PAI1 rs7242, and PAI1 rs1050813) (Figure 3A) and performed a series of cell-based assays, including cell proliferation, colony forming ability, migration, apoptosis, and cell–cell contact inhibition assays using these stable transfectants. We selected monoclonal stable transfectants that show similar levels of PAI1 mRNA and protein expression.

No significant differences regarding cell proliferation, colony forming ability, or migration were noted in either the UROtsa or 5637 cell line series (Appendix A).

To evaluate apoptosis, Annexin V and caspase 3/7 assays were performed 24 h after cisplatin treatment. Compared to C125E01 (control), UROtsa cells transfected with PAI1 WT-1H6, -2D7, rs7242-1C9, -2F10, and rs1050813 had significantly less Annexin V luminescence (29.9%, 46.4%, 10.8%, 3.0%, and 71.7%, respectively). Caspase 3/7 luminescence was also significantly decreased in UROtsa cells (33.3%, 26.3%, 28.6%, 39.2%, and 83.6%, respectively) (Figure 3B). These results demonstrate that PAI1 WT is involved in anti-apoptosis and the rs7242 transcript can strengthen the anti-apoptotic effect. In the 5637 cell series, PAI1 WT, SNPs rs7242 and rs1050813 all had lower Annexin V luminescence signals compared to the control (13.6%, 20.5%, and 8.9%, respectively), again supporting the idea that PAI1 expression is associated with an anti-apoptotic effect; however, there was no difference observed between WT and SNPs in this cell series, perhaps because PAI1 WT alone almost completely inhibited apoptosis induced by cisplatin. Conversely, caspase 3/7 luminescence was unchanged across the 5637 cell line series (113.8%, 120.1%, and 97.7%, respectively) (Figure 3B), suggesting that the apoptosis pathway in 5637 is not caspase 3/7-dependent.

### 2.8. Claspin Levels Are Associated with Apoptosis in PAI1-Expressing Cells, Specifically Cell Overexpressing PAI1 rs7242

To investigate how PAI1 might inhibit apoptosis, we used an array (Human Apoptosis Array Kit; ARY 009, Promega, Madison, WI, USA) which monitors 35 apoptosis-related proteins. Here, we compared the array data from cells treated with, or without, cisplatin treatment for 24 h. After cisplatin exposure, the level of cleaved-caspase 3 was induced in control UROtsa (empty vector) cells, contrary to UROtsa cells expressing PAI1 WT, rs7242, or rs1050813 which had reduced levels of cleaved-caspase 3. The cisplatin-induced levels of cleaved caspase-3 was nearly identical in 5637 control (empty vector), WT PAI1, rs7242, or rs1050813 expressing cells. The level of claspin (CLSPN), an essential upstream regulator of checkpoint kinase 1 that triggers a checkpoint arrest of the cell cycle, was much higher (fold change; 3.62) in rs1050813-expressing cells than in control UROtsa cells. In addition, CLSPN was further induced after cisplatin treatment in the PAI1 WT or rs7242 UROtsa cells compared to control UROtsa cells (fold change; 3.82 and 3.34, respectively) (Figure 3C). CLSPN levels in 5637 cells expressing PAI1 WT, rs7242, or rs1050813 were also higher than those in 5637 control cells (fold change; 1.90, 2.22, and 2.25, respectively), and CLSPN levels also increased after cisplatin treatment in PAI1 WT, rs7242, or rs1050813 compared to 5637 control cells (fold change; 2.01, 1.83, and 1.74, respectively) (Figure 3C). As CLSPN is known to be a main regulator of the ATR-CHEK1 cell cycle pathway after DNA damage [24], we evaluated the mRNA levels of ATR, CLSPN, and CHEK1 in cell treated with cisplatin. The mRNA expression levels of ATR, CLSPN, and CHEK1 were all increased after cisplatin treatment in all monoclonal cell clones in both UROtsa and 5637 monoclonal cell lines (Figure 3D). CLSPN is thought to act as a tumor suppressor via ATR-CHEK1 pathway activation and DNA repair [25], but reports have also shown that CLSPN can support tumor survival and is associated with poor prognosis in various types of cancers [26,27,28]. Given the diverse functions of CLSPN, we considered that the balance of CLSPN expression could vary between benign and cancer cells, which thus dictates differential function. To address this hypothesis, we silenced CLSPN expression by siRNA and evaluated cell survival and apoptosis in UROtsa (benign) and 5637 (urothelial carcinoma) cells (Figure 3E). The depletion of CLSPN in UROtsa cells resulted in significantly better cell survival (*p* = 0.0032) after cisplatin treatment compared to the negative control (scrambled siRNA), whereas in 5637 cells, CLSPN reduction resulted in worse post-treatment survival (*p* = 0.0004) (Figure 3F). Likewise, apoptosis was significantly suppressed (*p* = 0.0006) in CLSPN-depleted UROtsa cells compared to the control, while apoptosis was significantly augmented in CLSPN-depleted 5637 cells (*p* = 0.0008) (Figure 3G). These results suggest that the balance of CLSPN expression in benign and cancer cell types can influence the resistance or sensitization to apoptotic stimuli.
Figure 3PAI1 rs7242 is involved in the resistance to apoptosis induced by cisplatin. (**A**) Protein and mRNA expression of PAI1 clones in stable transfectants in UROtsa and 5637 cell lines. (**B**) Apoptosis assays using RealTime-Glo™ Annexin V Apoptosis and Caspase-Glo^®^ 3/7 Assays. (**C**) Profiling the expression of multiple proteins associated with apoptosis induced by cisplatin using a membrane-based sandwich immunoassay. (**D**) mRNA expression levels of ATR-CLSPN-CHEK1 pathway. (**E**) Silencing CLSPN by siRNA in parental UROtsa and 5637 cell lines. (**F**) Cell survival assay after cisplatin treatment. (**G**) Apoptosis assay after cisplatin treatment. Bars represent SE. ** *p* < 0.01; *** *p* < 0.001, **** *p* < 0. 0001.
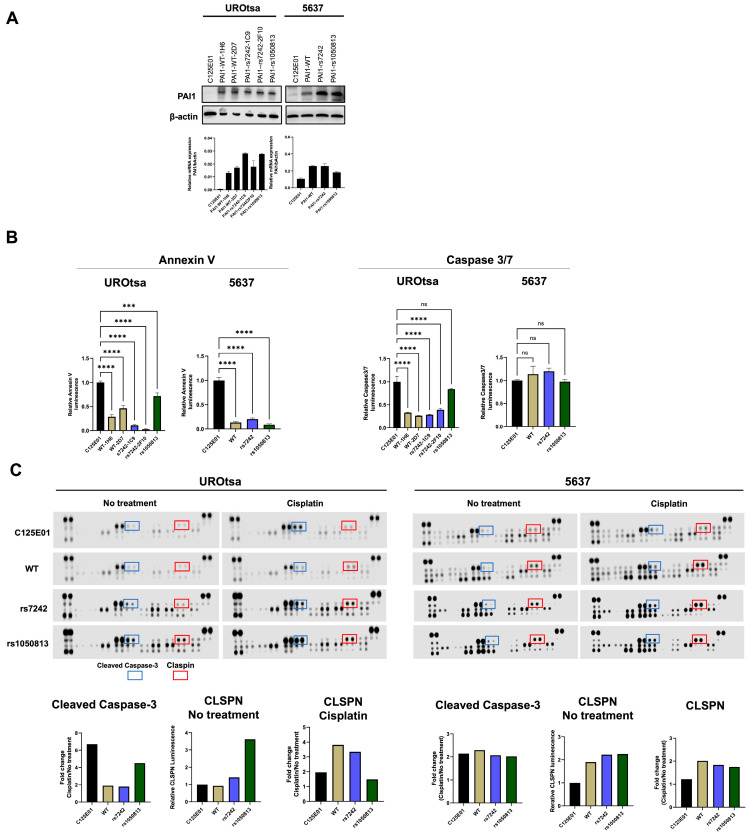

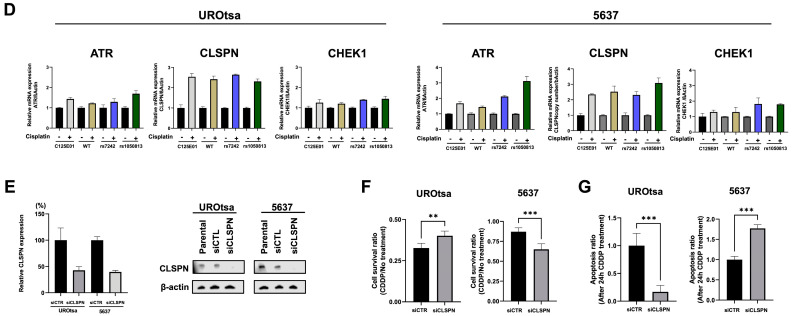


### 2.9. Cells Expressing PAI1 rs1050813 Lose Contact Inhibition of Proliferation

To evaluate the effect of contact inhibition on proliferation, the number of BrdU-positive cells at confluency was measured. The BrdU-positive cell rate for rs1050813 UROtsa and 5637 cells was significantly higher (*p* < 0.001, *p* < 0.01, respectively) than that of control, PAI1 WT, or rs7242-expressing cells (Figure 4A), indicating that rs1050813 is associated with a loss of contact inhibition of proliferation. Contact inhibition in cancer cells is in part regulated by YAP/TAZ [29], whereby YAP localizes to the nucleus in growing cells and to the cytoplasm at tissue culture confluence [30]. To examine this, we analyzed YAP protein expression in the nuclear and cytoplasmic fractions of UROtsa and 5637 cells. First, YAP protein was detected only in cytoplasmic fractions of UROtsa and 5637 cells. In UROtsa, PAI1 expressing cells (WT, rs7242 and rs1050813) had lower cytoplasmic YAP protein levels when they are low confluent (~30% of dish space was occupied), while there was no difference in high confluent cells (~100% of dish space was occupied) (Figure 4B). On the other hand, 5637 cells with high PAI1 (WT, rs7242 and rs1050813) showed higher cytoplasmic YAP expression than empty vector control when they are low and high confluency (Figure 4C). The results suggested that YAP is not involved in rs1050813-associated contact inhibition. As contact inhibition is also regulated by p53/Rb [31], we also evaluated p53 and Rb protein levels. In UROtsa rs1050813 cells, high Rb protein in sub-confluent cells was reduced at confluency, but no difference was observed in nuclear p53 levels in any of the cell lines (Figure 4B). The results suggested that rs1050813-associated contact inhibition may be mediated by the Rb pathway in UROtsa cells. Of note, 5637 cells are Rb-deficient, so rs1050813-associated contact inhibition in these cells must involve alternative pathways.

### 2.10. Bladder Cancer Patients with a High Expression of Both PAI1 and CLSPN Have Poor Disease-Free Survival

To investigate the relationship between PAI1 and CLSPN in a BCa patient population, we interrogated dataset GSE87304 available from the GEO database [32]. While PAI1 expression was not associated (*p* = 0.11) with disease-free survival in this cohort, high CLSPN expression was (*p* = 0.031) (Figure 5A). In addition, when PAI1 and CLSPN data were combined, patients with both high PAI1 and high CLSPN expression had significantly lower disease-free survival times (*p* = 0.011) compared to those with low/low expression. The association of PAI1 and CLSPN expression was also significantly (*p* = 0.049) associated with overall survival (Figure 5B). These results suggest that PAI1 and CLSPN may act synergistically to influence tumor growth and proliferation.

## 3. Discussion

In this study, 12 human urothelial cell lines, 92 case/control urine cell pellet samples (Cohort 1), 73 BCa tumors (Cohort 2), and 495 BCa tumors (Cohort 3) were examined for alterations in *PAI1* gene sequence. *PAI1* genetic alterations were detected in nine of the cell lines (75%). Point mutations identified in exon 2 were non-synonymous (ns) SNPs located in the signal peptide (nsSNP A15T rs6092; nsSNP V17I rs6090), and point mutations (rs11178; rs7242) and a 9-bp insertion (rs41423845) were identified in the 3′ UTR of exon 9. Additional SNPs were identified in the patient cohorts, namely, rs1050813, rs1050955, 1,050,766 (Cohort 1) and rs7242 and rs1050813 (Cohort 2), all of which are in *PAI1* exon 9. Based on prevalence and the association with clinical outcome, the 3′ UTR SNPs rs7242 and rs1050813 were selected as the focus for further analysis.

To further investigate the association of these SNPs with survival, we analyzed a combined, diverse patient cohort comprised of 568 tumor tissue samples from 310 Asians, 200 Caucasians, and 58 others. The incidence of rs7242 (somatic) was higher in Asians at 76.3% (209/274 cases; 36 cases were removed because of sequence data quality) relative to Caucasians at 65.5% (91/139 cases; 61 cases were removed). Conversely, the incidence of rs1050813 (germline) was higher in Caucasians at 40.9% (65/159 cases; 41 cases were removed) relative to Asians at 5.9% (18/306 cases; 4 cases were removed). Kaplan–Meier analyses restricted to the Asian patients within the cohort found no difference in overall survival (OS) and recurrence-free survival (RFS) between *PAI1* WT or mutated SNPs rs7242 and/or rs1050813 (Appendix A). In the Caucasian subject subset, patients harboring either rs7242 or rs1050813 had a worse OS and RFS (*p* < 0.03 and *p* < 0.03, respectively), whereas neither rs7242 alone nor rs1050813 alone had significant differences in OS and RFS (Figure 1).

Smoking is one of the major risk factors in the development of BCa in industrialized countries, and smoking rates are notoriously higher in Asia than in Northern America [33]; however, the overall incidence of BCa is lower in Asia than it is in Northern America [34]. Therefore, it is possible that this phenomenon is associated with the low incidence of rs1050813 (germline SNP) in Asians, i.e., the incidences of rs1050813 in Caucasian and Asian are 40.9% vs. 5.9%, respectively.

SNPs are genetic changes, which occur in both the coding and noncoding regions of genes. Due to the redundancy of the genetic code, some alterations do not change the encoded amino acid sequence and are considered synonymous mutations, including those in the 3′ UTR [35]. However, synonymous SNPs can affect gene function. Synonymous SNPs have been shown to affect mRNA secondary structure, splicing and stability in addition to altering translation kinetics, protein folding and therefore protein function [35,36,37]. Thus, we investigated whether the observed *PAI1* genetic alterations affected mRNA expression, alternative splicing, miRNA binding or mRNA secondary structure.

In human specimens (Cohort 2), no variation in *PAI1* mRNA alternative splicing was detected. miRNAs are small (18–22 nt), non-coding RNAs that can regulate target mRNA processing through stability or by inhibiting translation [38,39,40]. We used bioinformatic prediction programs to predict miRNA-mRNA interactions at the 3′ UTR of *PAI-1.* Analysis using STarMir revealed that two of the genetic alterations observed in *PAI1* with an association with BCa recurrence (rs1050813 and rs1050955) were potential targets for miRNA binding. Specifically, analysis showed that miR-143-3p would have a higher binding affinity for the WT compared to rs1050813, likely resulting in lower *PAI1* expression levels. Similarly, the WT transcript was predicted to be a more probable target for miR-548ah-5p compared to rs1050955. A second miRNA, miR-5689, was also predicted to bind to the WT and not to rs1050955 transcripts. The activity of the miR-143 family has been linked to a number of different cancer types, including BCa [41,42,43,44,45]. Notably, Villadsen et al. previously reported that miR-143 and miR-145 can directly target the *PAI1* 3’ UTR, leading to reduced PAI1 mRNA and protein levels [46]. Another study showed that miR-421 and miR-30c could directly interact with *PAI1* mRNA at the rs1050955 location, resulting in the inhibition of PAI1 protein expression in human umbilical vein endothelial cells (HUVEC) [38], implicating a role for rs1050955 in the regulation of PAI1 expression. It has also been shown that SNPs can affect mRNA folding [47], and variations in mRNA secondary structure and stability may impact protein translation as discussed previously [37,48]. Using RNAfold, differences in mRNA stability and folding were predicted for the following alterations in *PAI1* 3′ UTR, rs7242 and rs1050813, because only these two SNPs showed correlation to progression- and recurrence-free survival in Cohort 2. If a synonymous SNP results in a different mRNA structure, ribosome binding and translation rate would change [49]. Thus, translational pausing may affect the final conformation of the protein. However, no difference in PAI1 protein levels was observed between WT and SNPs in UROtsa transiently transfected with *PAI1* constructs, assuming that *PAI1* SNPs may not affect protein expression levels. Therefore, it is suggested that there may be a difference in protein functions.

To further study how *PAI1* SNPs are associated with OS and RFS, we generated a series of stably transfected cell lines that expressed PAI1 WT, rs7242 and rs1050813 mRNA, and monitored the effects using a series of cell-based assays. We selected UROtsa (benign) and 5637 (urothelial carcinoma) cells because their PAI1 expression levels are quite low. First, we performed gene expression microarray to explore the downstream pathway of synonymous SNPs. The gene expression microarray identified 79 common pathways, which included cell proliferation, migration, and apoptosis-related. Next, although there were no significant differences among PAI1 genotypes (PAI1 WT, rs7242 and rs1050813) with respect to cell proliferation, colony forming ability, or migration in either UROtsa or 5637 cell series, PAI1 SNPs rs7242 and rs1050813 augmented the effect of PAI1 on apoptosis and cell–cell contact inhibition, respectively. Furthermore, the overexpression of PAI1 WT inhibited cisplatin-induced apoptosis and the expression of rs7242 enhanced the anti-apoptotic effect in UROtsa cells. However, in 5637 cells, PAI1 expression was associated with an anti-apoptotic effect, but there was no difference between WT and SNP-transcript expression; PAI1 WT almost completely inhibited cisplatin-induced apoptosis, a response that appears to be caspase 3/7-independent. Because there is little difference in PAI1 mRNA and protein levels among clones, these results suggest that the differences are due to the PAI1 genotype, not expression levels. Protein array analyses identified CLSPN as a potential factor in rs7242-associated apoptosis inhibition. CLSPN is known to be a multifunctional protein, acting as a tumor suppressor in some cases and as a tumor progressor associated with poor prognosis in different types of cancers [26,50]. We hypothesized that this duality of CLSPN function may be regulated by PAI1 status. To test this, we silenced the CLSPN expression by siRNA in both UROtsa and 5637 cells and monitored the effect on cell survival and apoptosis. Interestingly, CLSPN knockdown inhibited apoptosis and increased cell survival in UROtsa cells (benign), while CLSPN knockdown enhanced apoptosis and decreased cell survival in 5637 cells (urothelial carcinoma). The function of CLSPN in benign versus cancerous cells will depend on multiple factors, including PAI1, but the results suggest that rs7242-associated apoptosis inhibition is mediated in part by CLSPN in UROtsa. We also found that rs1050813 clones did not arrest growth in culture upon contact with other cells, i.e., as the 2D culture becomes confluent. It has been reported that this contact inhibition of proliferation in cancer cells is regulated by the YAP/TAZ [29] and p53/Rb pathways [31]. As YAP was not involved in rs1050813-associated contact inhibition, we analyzed p53 and Rb protein levels. In UROtsa cells, rs1050813-associated contact inhibition was associated with reduced Rb protein levels; however, the observed rs1050813-associated contact inhibition in 5637 must involve other pathways because 5637 is Rb-deficient and p53 levels were not associated with contact inhibition.

According to the cell-based assays, including apoptosis array or siRNA mediated knock down assay, CLSPN was identified as an important downstream molecule of PAI1 overexpression or SNP mutations. In fact, survival and recurrence analysis by GEO data (GSE87304) showed the positive correlation between PAI1 and CLSPN. These findings suggest that PAI1 and CLSPN may synergistically work to promote cancer cell proliferation.

The present study has some limitations that are worth mentioning. First is its retrospective design and the inherent selection bias. Second, mainly FFPE tissues were available to test, and thus this study was limited to DNA profiling. It is feasible that freshly frozen tissue samples may provide different results.

## 4. Materials and Methods

### 4.1. Bladder Cell Lines

Twelve bladder cell lines, consisting of 10 urothelial carcinoma, 1 papilloma (RT4) and 1 benign bladder cell line (UROtsa), were available for the prescribed studies. Cell lines were obtained from the Pathology Core of the Bladder Cancer SPORE at MD Anderson Cancer Center (RT112, RT4, 5637, UM-UC-1, UM-UC-3, UM-UC-13, UM-UC-14, TCCSUP, 253-J, and 253J-BV cells), T24 cells from American Type Culture Collection (Manassas, VA, USA), and the UROtsa benign human bladder cell line from Dr. Donald Sens of the University of North Dakota School of Medicine, Grand Forks, ND. Prior to experimentation, the cell lines were authenticated by the Genetic Resources Core Facility at Johns Hopkins (Baltimore, MD, USA).

### 4.2. Patient Samples

The study was performed after approval by the Western Institutional Review Board (IRB # 20141019) under a request of waiver of consent on archived pathologic specimens. After the IRB approval, samples from the following cohorts were made available for analysis.

### 4.3. Cohort 1—Screening Cohort

Ninety-two DNA samples (case = 69; and control = 23) extracted from urine cell pellets from the Early Detection Research Network (EDRN) (NCI, Bethesda, MD, USA) were made available.

### 4.4. Cohort 2—Discovery Cohort

Seventy-three fresh frozen primary bladder tumors with their matched buffy coat from peripheral blood samples obtained at the time of surgery were made available. Demographic characteristics of the study population are summarized in Table 1. Primary tumors consisted of a set of 4 low grade tumors, 66 high grade tumors, 26 non-muscle invasive tumors (Ta, T1, and Tis) and 46 muscle invasive tumors (>T1). Tumor tissues and buffy coat samples were stored at −80 °C until DNA and RNA extraction. Median follow-up of the cohort was 23.4 months.

### 4.5. Cohort 3—Replication Cohort

Formalin fixed paraffin embedded (FFPE) bladder tissues from patients with a diagnosis of urothelial carcinoma were identified at Mayo Clinic (*n* = 153), Chinese University of Hong Kong (*n* = 92), Chia-Yi Christian Hospital, Taiwan (*n* = 64), Nara Medical University (*n* = 49), Kyoto University (*n* = 58), and University of Hawaii (*n* = 79), and shipped to University of Hawaii for testing. Consecutive cases were chosen based on (1) the confirmed presence of BCa within the block and (2) the availability of the paraffin block. Medical records were reviewed for the following information: age, gender, race, tumor histology, tumor grade, tumor stage, tobacco history, and clinical outcomes. Demographic characteristics of the study population are summarized in Table 1. Median follow-up of the cohort was 59.1 months.

### 4.6. Clinical Outcomes

Pertinent information on tumor recurrence, progression and overall survival were obtained from each site. Specifically, the following definitions were used to ensure consistency of the data. In NMIBC, a recurrence was defined as a new tumor noted on subsequent cystoscopy, which was shown by biopsy to be urothelial carcinoma, a progression was a patient with Ta going to T1, T1/Tis going to T2, or low grade going to high grade. In MIBC, a recurrence after cystectomy was the identification of a growing pelvic mass, enlarging lymph nodes, bone disease, or any other metastatic disease, a progression was a patient with tumor confined to the bladder who was noted to have ‘recurrent’ disease or a patient treated with chemotherapy or any other systemic agent(s), in which the disease showed an initial response, but then the disease showed increased burden; death was noted as dead of disease (DOD, i.e., of bladder cancer) or dead of other causes.

### 4.7. Mutation Analysis of the PAI1 Gene

Genomic DNA and total RNA were extracted from bladder cell lines and Cohort 2 using AllPrep DNA/RNA Kit (Qiagen, Germantown, MD, USA) and cohort 3 using QIAamp DNA FFPE Tissue Kit (Qiagen). Extracted DNA from Cohort 1 was provided by EDRN. PCR primers were designed to amplify the promoter and each exon in *PAI1*. DNA were used for PCR amplification. PCR reactions were carried out in a total volume of 25 µL containing genomic DNA template, 0.4 µM of each PCR primer, and GoTaq G2 Hot Start Green Master Mix (Promega, Madison, WI, USA), except for *PAI1* exon 9. Due to the length of exon 9′s PCR products, AccuStart II GelTrack PCR Super Mix (Quanta BioSciences, Inc., Gaithersburg, MD, USA) was used. Forty cycles of 30 s at 94 °C, 30 s at 60 °C, and 30–120 s at 72 °C (per intended PCR product) were performed in a programmable thermal cycler (Bio-Rad, Hercules, CA, USA). PCR products were checked on a 1.5% agarose gel, followed by PCR purification and bidirectional Sanger sequencing (Psomagen, Rockville, MD, USA). Sequence analysis was performed using Geneious, (Geneious version 8, “http://www.geneious.com (accessed on 5 August, 2015)”, [51]) and detected variants were confirmed against a RefSeq genomic DNA sequence (PAI1: NG_013213.1). All genetic alterations identified by Geneious were compared to the NCBI SNP database (dbSNP).

### 4.8. mRNA Expression Analysis

Total RNA from Cohort 3 was reverse-transcribed using qScript^TM^ Reverse Transcription Supermix for RT-qPCR (Bio-Rad, Hercules, CA, USA) according to the manufacturer’s instruction. cDNA was subjected to real-time PCR amplification of PAI1 transcripts as described in Appendix A.

### 4.9. Bioinformatic Analysis

TRANSFAC Database Match tool [51,52] was used to annotate potential transcription factor (TF) binding sites within the promoter. For secondary structure of the mRNA, the sequence was predicted using the RNAfold webserver (http://rna.tbi.univie.ac.at/cgi-bin/RNAfold.cgi (accessed on 3 September 2016)). We searched DIANA microT-CDS (“http://diana.imis.athena-innovation.gr/DianaTools/index.php?r=microT_CDS/index (accessed on 3 September 2016)”) for miRNA binding sites within mutated regions of both polymorphic sites, and confirmed our results using STarMiR (“http://sfold.wadsworth.org/cgi-bin/starmir.pl (accessed on 3 September 2016)”). In order to predict the functional consequences of mutations on protein sequence, we used four software programs: 1. SNPeffect4.0 (http://snpeffect.switchlab.org (accessed on 3 September 2016)), 2. Phyre2 (www.sbg.bio.ic.ac.uk/phyre2/ (accessed on 3 September 2016)) 3. SIFT (http://sift.jcvi.org/ (accessed on 3 September 2016)), and 4. PolyPhen2 (http://genetics.bwh.harvard.edu/pph/ (accessed on 3 September 2016)).

### 4.10. Expression of PAI1 Constructs in UROtsa and 5637 Cell Lines

Transfection of PAI1 constructs containing the rs7242 or rs1050813 SNPs were performed in UROtsa and 5637 cell lines. Transient transfectants were used for microarray gene expression analysis, while stable transfectants were used for a series of cell-based assays. Stable transfectants were selected as described in Appendix A.

### 4.11. Microarray Gene Expression Analysis

Microarray assay (Human HT-12 v4.0 Expression Beadchip, Illumina, San Diego, CA, USA) and the process of raw scanned data were performed by Psomagen. Detailed methods are described in Appendix A.

### 4.12. mRNA Expression of PAI1, CLSPN, ATR, and CHEK1 Genes

Total RNA was extracted from stable transfectants of UROtsa and 5637 cells using the RNeasy Mini Kit (Qiagen) according to the manufacturer’s protocols, and complementary DNA was synthesized from 1 μg of RNA using the ReverTra Ace qPCR RT Kit (TOYOBO, Osaka, Japan). qRT-PCR was performed using SYBR Green Master Mix (Applied Biosystems, Foster City, CA, USA) and the QuantStudio Real-time PCR system (Applied Biosystems) as described in Appendix A.

### 4.13. Cellular and Molecular Assays

A series of cellular and molecular assays, such as cell-based assays and mechanistic assays were performed as described in Appendix A.

### 4.14. Publicly Available Cohorts for Validation of SNP Associations

The Gene Expression Omnibus cohort, GSE87304, includes 303 MIBC patients treated with neoadjuvant chemotherapy with the primary outcome of recurrence-free survival was accessed [32].

### 4.15. Statistical Analysis

Fisher’s exact test was used to assess the differences of *PAI1* alterations in bladder cell lines and the case/control cohort (Cohort 1). Wilcoxon rank sum test was used to determine a correlation based on the number of genetic alterations in either SNP. One-way ANOVA or student’s t-test was used to assess the presence of genetic alterations on gene expression. The correlation between SNPs and clinical outcomes including recurrence, progression, and overall survival were evaluated using multivariate Cox statistical analysis adjusting for age, stage, and grade as covariates. Kaplan–Meier curves were constructed for recurrence-free, disease-free survival, and overall survivals and the survival differences by SNP genotypes were compared using the log-rand test. The alterations were entered into a stepwise Cox proportional hazards model, which selected the best predictors. All variables were entered simultaneously in the final model and median years to recurrence were estimated separately in a parametric model. In all models, age at diagnosis was included as a covariate. SAS V9.4 (SAS Institute Inc., Cary, NC, USA) and GraphPad Prism 7.0 (GraphPad Software Inc., La Jolla, CA, USA) were used to perform statistical analyses. A *p*-value of <0.05 was considered statistically significant.

## 5. Conclusions

In conclusion, we demonstrated that *PAI1* genetic alterations occurred in BCa with SNPs rs7242 and rs1050813 being significantly associated with OS and RFS in Caucasians, but not in Asians. In vitro studies demonstrated that rs7242 strengthened anti-apoptotic function of PAI1 WT by mediating CLSPN and rs1050813 was associated with the contact inhibition of proliferation probably mediated by the combination of many pathways including Rb. Taken together, the results suggest that *PAI1* SNPs rs7242 and rs1050813 may be associated with BCa OS and RFS via mediating inhibition of apoptosis and losing contact inhibition of proliferation. The findings from this study will eventually lead to the development of new biomarkers for prognosis, though additional studies are needed to confirm these intriguing results.

## Figures and Tables

**Figure 2 ijms-24-04943-f002:**
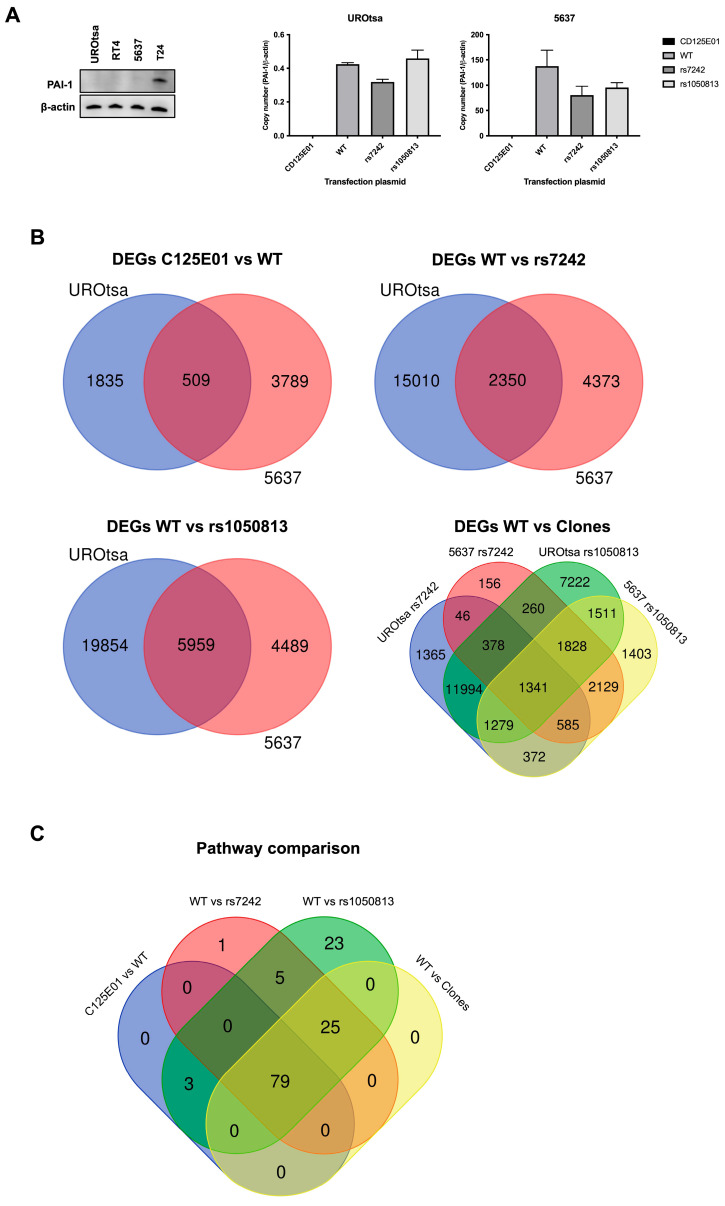
The effects of PAI1 SNPs in its downstream pathways. (**A**) PAI1 protein expression in urothelial cell lines. Plasmids for C125E01, PAI1 WT, PAI1 rs7242, and PAI1 rs1050813 were transiently transfected into UROtsa and 5637 cells which show no or low PAI1 expression. (**B**) Venn diagram showing the overlapping genes in DEGs C125E01 vs. WT, DEGs WT vs. rs7242, DEGs WT vs. rs1050813, and DEGs WT vs. both clones between UROtsa and 5637 cells. (**C**) Venn diagram representing the overlap of pathways identified based on the overlapping genes in Figure 2B.

**Figure 4 ijms-24-04943-f004:**
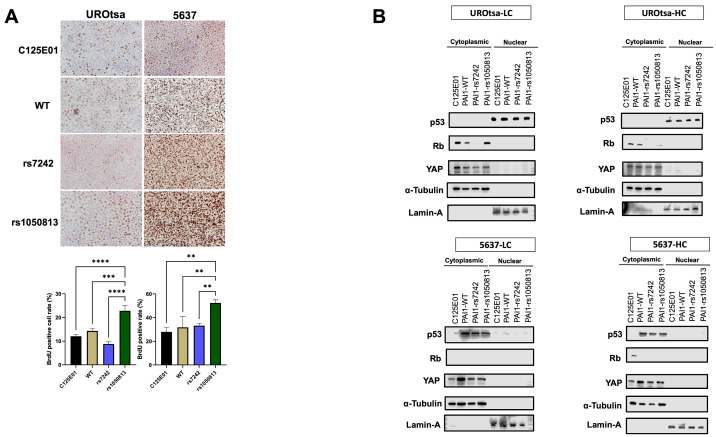
PAI1 rs1050813 aborts cell–cell contact inhibition. (**A**) Evaluation of cell proliferation in high-cell density/confluency condition by ICC for BrdU. (**B**) Protein levels of p53, Rb, and YAP in subcellular fractions. Bars represent SE. ** *p* < 0.01; *** *p* < 0.001, **** *p* < 0. 0001.

**Figure 5 ijms-24-04943-f005:**
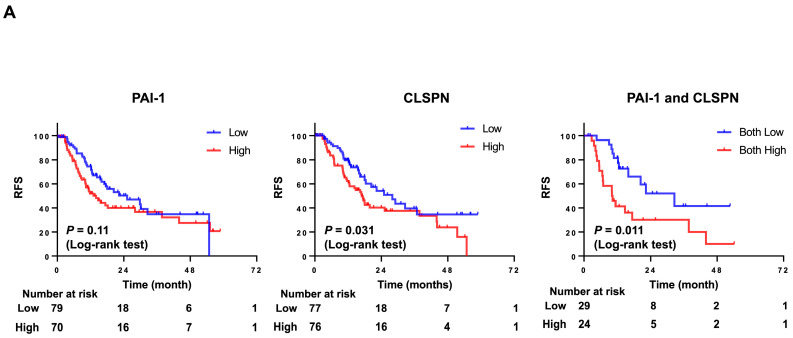
Kaplan–Meier survival curves for high vs. low expression of PAI1 alone, CLSPN alone, and the combination of PAI1 and CLSPN in GEO cohort (GSE87304). (**A**) RFS and (**B**) OS.

**Table 1 ijms-24-04943-t001:** Demographic characteristics among patients with bladder cancer.

Variable		Discovery (*n* = 73) *n* (%)	Replication (*n* = 495) *n* (%)	Pooled (*n* = 568) *n* (%)	P ^A^
Race	Asian	0 (0%)	310 (63%)	317 (55%)	**<0.0001**
	White	56 (77%)	144 (29%)	212 (37%)	
	Other	17 (23%)	41 (8%)	51 (9%)	
Gender	Female	14 (19%)	86 (17%)	115 (20%)	1.00
	Male	59 (81%)	409 (83%)	465 (80%)	
Age	Under 65	20 (27%)	114 (23%)	138 (24%)	**0.006**
	65–75	37 (51%)	179 (36%)	221 (38%)	
	Over 75	16 (22%)	202 (40%)	221 (38%)	
Tobacco Use	No	17 (25%)	344 (69%)	323 (63%)	**<0.0001**
	Yes	52 (75%)	151 (31%)	192 (37%)	
Grade	Low	4 (6%)	97 (20%)	101 (18%)	**0.003**
	High	66 (94%)	372 (75%)	438 (77%)	
	Unknown	3 (4%)	26 (5%)	29 (5%)	
Stage	T0 or Ta	8 (11%)	139 (28%)	147 (26%)	**0.0009**
	Tis or T1	18 (25%)	146 (29%)	164 (29%)	
	T2-T4	46 (63%)	210 (43%)	256 (45%)	
	Unknown	1 (1%)	0 (0%)	1 (0.2%)	
Recurrence	No	53 (73%)	228 (46%)	281 (50%)	**<0.0001**
	Yes	20 (27%)	260 (53%)	280 (49%)	
	Unknown	0 (0%)	7 (1%)	7 (1%)	
Progression	No	53 (73%)	324 (66%)	377 (66%)	0.35
	Yes	20 (27%)	164 (33%)	184 (33%)	
	Unknown	0 (0%)	7 (1%)	7 (1%)	
Invasive	No	26 (36%)	290 (59%)	316 (56%)	**0.0005**
	Yes	46 (64%)	205 (41%)	251 (44%)	
	Unknown	1 (1%)	0 (0%)	1 (0.2%)	
Dead	No	62 (85%)	233 (48%)	295 (53%)	**<0.0001**
	Yes	11 (15%)	255 (52%)	266 (47%)	
	Unknown	1 (1%)	7 (1%)	8 (1%)	
Follow-Up ^B^		23.4 (18.6–28.2)	59.1 (46.8–71.4)	54.6 (43.8–65.4)	

^A^*p* value was calculated by Fisher’s exact test. Bold indicates that the data is significantly different (P<0.05). ^B^ Mean follow-up time in months with 95% CIs shown.

**Table 2 ijms-24-04943-t002:** Frequencies of common genetic alterations found in *PAI-1* genes.

	Total		Recurrence				Progression				Overall		
	N	*n* (%)	HR (95% CI) ^A^	P ^A^	MST ^B^	*n* (%)	HR (95% CI)	P	MST	*n* (%)	HR (95% CI)	P	MST
**rs7242**													
Discovery													
TT	21	6 (29%)	1.00			5 (24%)	1.00			1 (5%)	1.00		
TG	26	2 (8%)	0.72 (0.19–2.70)	0.63		3 (12%)	0.82 (0.21–3.22)	0.78		4 (15%)	3.03 (0.33–28.08)	0.33	
GG	14	6 (43%)	2.16 (0.57–8.26)	0.26		6 (43%)	2.35 (0.59–9.27)	0.22		5 (36%)	5.16 (0.58–46.24)	0.14	
P-trend ^C^		0.52				0.29				**0.02**			
Replication													
TT	117	62 (53%)	1.00		22.5	34 (29%)	1.00		65.1	56 (48%)	1.00		81.8
TG	226	117 (52%)	0.95 (0.72–1.27)	0.75	23.1	79 (35%)	1.08 (0.79–1.49)	0.63	51.9	126 (56%)	1.07 (0.76–1.50)	0.72	74.8
GG	93	52 (56%)	1.17 (0.83–1.65)	0.36	15.1	34 (37%)	1.35 (0.91–2.00)	0.14	36.4	51 (55%)	1.47 (0.97–2.23)	0.07	47.8
P-trend		0.71				0.24				0.28			
Pooled													
TT	138	68 (49%)	1.00		24.9	39 (28%)	1.00		68.4	57 (41%)	1.00		88.4
TG	252	119 (47%)	0.98 (0.75–1.30)	0.91	25.8	82 (33%)	1.11 (0.81–1.51)	0.53	54.7	130 (52%)	1.13 (0.81–1.59)	0.47	75.8
GG	107	58 (54%)	1.26 (0.91–1.75)	0.17	16.3	40 (37%)	1.41 (0.97–2.06)	0.07	36.8	56 (52%)	1.59 (1.06–2.37)	**0.02**	49.7
P-trend		0.50				0.13				0.07			
**rs1050813**													
Discovery													
GG	51	14 (27%)	1.00			13 (25%)	1.00			7 (14%)	1.00		
GA	20	4 (20%)	0.56 (0.16–1.94)	0.36		5 (25%)	0.69 (0.20–2.42)	0.56		4 (20%)	1.23 (0.32–4.77)	0.77	
AA	1	1 (100%)				1 (100%)				0 (0%)			
P-trend		0.92				0.52				0.67			
Replication													
GG	396	206 (52%)	1.00		21.6	121 (31%)	1.00		50.4	200 (51%)	1.00		70.5
GA	72	40 (56%)	0.90 (0.64–1.27)	0.55	25.0	35 (49%)	1.17 (0.81–1.67)	0.41	38.8	42 (58%)	1.22 (0.84–1.76)	0.29	53.0
AA	7	6 (86%)	1.21 (0.57–2.60)	0.62	13.2	4 (57%)	1.22 (0.50–2.99)	0.66	36.0	4 (57%)	1.19 (0.44–3.24)	0.73	53.8
P-trend		0.16				**0.001**				0.23			
Pooled													
GG	447	220 (49%)	1.00		22.6	134 (30%)	1.00		51.9	207 (46%)	1.00		72.8
GA	92	44 (48%)	0.84 (0.60–1.16)	0.28	31.7	40 (43%)	1.07 (0.76–1.51)	0.71	47.0	46 (50%)	1.16 (0.81–1.65)	0.42	60.4
AA	8	7 (88%)	1.40 (0.69–2.87)	0.35	12.2	5 (63%)	1.52 (0.67–3.44)	0.32	23.4	4 (50%)	1.25 (0.46–3.38)	0.66	53.0
P-trend		0.38				**0.002**				0.52			

^A^ HRs, 95% CIs, P values, and MSTs were calculated with multivariate Cox models adjusting for age, stage, and grade as covariates. Bold indicates that the data is significantly different (*p* < 0.05). ^B^ MST is median survival time in months. In the Discovery cohort, median survival was not reached. ^C^ P-trend is the *p*-value from the Cochran–Mantel–Haenszel test for the change in percentage trend.

## Data Availability

The datasets generated and/or analyzed during the current study are available in the Gene Expression Omnibus (GEO) repository, GSE224309.

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
