# Peer review of "Association of SNPs in the PAI1 Gene with Disease Recurrence and Clinical Outcome in Bladder Cancer"

_ijms, 2023, doi:10.3390/ijms24054943_

Round 1

Reviewer 1 Report

The manuscript offers new insights on the significance of specific Single Nucleotide Polymorphisms in the gene of Plasminogen Activation Inhibitor 1, with special reference to the Overall Survival and Recurrence Free Survival rate in patients with bladder cancer. It offers experimental data explaining molecular mechanisms through which SNPs in the 3’UTR of PAI1 may influence apoptosis and cell-cell contact inhibition – both highly relevant in the progression of any type of cancer.

Article is well-written, with clear objectives and logical sequence of experiments, involving numerous up-to-date experimental methods, as well as bioinformatics and statistical software tools. Results and conclusions are in line with the main topic of the Special Issue.

Obtained data is impressive, of good quality, derived from a adequate number of experiments and cohorts of patients enrolled in the analyses.

Therefore, I recommend the manuscript for publication after resolving minor issues of technical nature.

Minor Issues

Figures in the manuscript are of lower quality/resolution compared to the figures in the supplement. Please correct.

Line 580:             full terms for abbreviations OS and RFS should be added in the text, and abbreviations in the legend of Figure 1. It is a bit confusing since in the Figure 5. With Kaplan–Meier survival curves, different term is used. Is the “Disease-free survival” in figure 5 the same parameter as RFS used in Figure 1 and the text?

Line 588:             percentages should be in reverse order 5.9% vs 40.9%, respectively – on order to follow previous sentence. Now it is written: “the overall incidence of BCa incidence is lower in Asia than it is in Northern America [36]. Therefore, it is possible that this phenomenon is associated with the low incidence of rs1050813 (germline SNP) in Asians, i.e., 40.9% vs. 5.9%, respectively.

Lines 671-679                    In conclusion, translational significance of this research with reference to the development of new biomarkers for BCa should be acknowledged.

Supplemental Figure 8                  methods/reference should be given in the legend for Cell proliferation, Clonogenic assay, and Scratch assay

Author Response

Please find an attached file for the point-by-point response.

Reviewer 2 Report

In this manuscript the authors sought out to clarify the role of single nucleotide polymorphisms (SNPs) in PAI1 in the context of disease recurrence among bladder cancer pts.

Molecular and gene-related signature represent an exiting and rapidly emerging platform of interest both considering neaoadjuvant (doi: 10.1016/j.ajur.2021.05.001) and upfront RC plus with adjuvant treatments (doi: 10.1016/j.euo.2021.04.004). However, the lack of properly design external and prospective validation are currently hindering the stable introduction of such tools in the clinical daily practice. The authors should refer, discuss and compare their findings with the current evidence of molecular biomarkers platforms. In this scenario the effort of the authors is to be commended as the study is properly designed with a discovery, replication and pooled cohorts.

Beyond the promising findings of SNPs PAI1-related alterations, the authors should discuss and compare their findings with the already well established tools in disease prognosis prediction such as variant histology (doi: 10.1111/bju.15984), immunohistochemestry-based biomarkers (doi: 10.1016/j.urolonc.2021.10.010) and conventional laboratory-based parameters (doi: 10.1016/j.clgc.2023.01.008; doi: 10.1016/j.urolonc.2021.04.026; doi: 10.1007/s10147-021-01952-6). All these tools are easily accessible, ready-to-use, and cost-effective. The authors should refer, discuss and compare their findings with these evidence.

A paragraph about limitations (retrospective design) is highly recommended at the end of the Discussion paragraph.

Author Response

(The authors gave the same response as above.)

Round 2

Reviewer 2 Report

The authors provided a revised version of the manuscript. However, the proposed references were not discussed in the appropriate sections of the manuscript. Thus the manuscript need to be further developed considering the previous suggestions.

As previously proposed molecular and gene-related signature represent an exiting and rapidly emerging platform of interest both considering neaoadjuvant (doi: 10.1016/j.ajur.2021.05.001) and upfront RC plus with adjuvant treatments (doi: 10.1016/j.euo.2021.04.004). However, the lack of properly design external and prospective validation are currently hindering the stable introduction of such tools in the clinical daily practice. Beyond the promising findings of SNPs PAI1-related alterations, the authors should discuss the already well established tools in disease prognosis prediction such as variant histology (doi: 10.1111/bju.15984), immunohistochemestry-based biomarkers (doi: 10.1016/j.urolonc.2021.10.010) and conventional laboratory-based parameters (doi: 10.1016/j.clgc.2023.01.008; doi: 10.1016/j.urolonc.2021.04.026; doi: 10.1007/s10147-021-01952-6). All these tools are easily accessible, ready-to-use, and cost-effective. The authors should refer, discuss and compare their findings with these evidence.

Since the authors claimed that they have demonstrated that PAI1 genetic alterations occurred in BCa with SNPs rs7242 and rs1050813 being significantly associated with OS and RFS they should further discuss the previous evidence of the literature both in this very specific topic and considering the already well-established tools easy accessible in the clinical daily practice. Furthermore, in the era of both precision medicine and the careful attention to the related costs of procedures why these findings are so important compared to the previous literature? Please, develop this point  in the appropriate section of the manuscript.

Author Response

Thank you very much for your initial review and comments. We decided not to include the proposed references and the discussion in relation to the references, and explained why we do not think your suggestions suit our manuscript. Would you please provide us with the reason why you still think these references and discussions are necessary?